# Chi-Circ_0006511 Positively Regulates the Differentiation of Goat Intramuscular Adipocytes via Novel-miR-87/CD36 Axis

**DOI:** 10.3390/ijms232012295

**Published:** 2022-10-14

**Authors:** Xin Li, Hao Zhang, Yong Wang, Yanyan Li, Youli Wang, Jiangjiang Zhu, Yaqiu Lin

**Affiliations:** 1Key Laboratory of Qinghai-Tibetan Plateau Animal Genetic Resource Reservation and Utilization, Southwest Minzu University, Ministry of Education, Chengdu 610041, China; 2Key Laboratory of Qinghai-Tibetan Plateau Animal Genetic Resource Reservation and Exploitation of Sichuan Province, Southwest Minzu University, Chengdu 610041, China; 3College of Animal and Veterinary Sciences, Southwest Minzu University, Chengdu 610041, China

**Keywords:** goat, circRNA, adipocyte differentiation, novel-miR-87, CD36

## Abstract

Goats are an important livestock and goat meat is essential to local people. The intramuscular fat (IMF) content has a great influence on the quality of goat meat. The intramuscular preadipocytes differentiation is closely related to the IMF deposition; however, its potential regulatory mechanisms remain unclear. CircRNAs were revealed to be involved in multiple biological progressions. In this study, we took primary goat intramuscular preadipocyte (GIMPA) as the study model to verify the function and mechanism of chi-circ_0006511, which was abundant and up-regulated in mature adipocytes (GIMA). The results showed that the expression level of chi-circ_0006511 gradually increased in the early stage of GIMPA differentiation, and chi-circ_0006511 was confirmed to promote GIMPA lipid droplets aggregation and up-regulate the adipogenic differentiation determinants, further promoting GIMPA differentiation. Mechanistically, chi-circ_0006511 exerts its function by sponging novel-miR-87, thereby regulating the expression of CD36. The results from this study provided novel significant information to better understand the molecular regulatory mechanism of intramuscular preadipocytes differentiation, thereby providing a new reference for the intramuscular fat adipogenesis in goats.

## 1. Introduction

Goats are the most widespread of all domesticated ruminants due to their extreme adaptability [1]. Consumers are increasingly interested in goat meat as these animals have a small amount of subcutaneous and intramuscular fat [2]. The potential of goats to produce high-quality meat is mainly reflected in their healthy fats, low-calorie intramuscular fats, saturated fats, and especially their high proportions of unsaturated and saturated fatty acids [3].In addition, appearance, tenderness, taste and juiciness are important categories that influence consumer acceptance of goat meat [4,5].The content of intramuscular fat (IMF) is one of the key factors in determining the tenderness and juiciness of meat, and also greatly affects the flavor [6,7]. The study of goat IMF deposition is of great significance for improving meat quality and breeding. While IMF deposition was mainly determined by the hyperplasia and hypertrophy of intramuscular adipocytes (IMA) [8], adipocyte differentiation is an important part of IMF deposition, which is an extremely complex physiological process that is tightly regulated by multiple transcription factors and noncoding RNAs(ncRNA) [9,10,11,12]. PPARγ and C/EBPα play key roles in adipogenic differentiation as classic adipogenic differentiation marker genes [13,14,15]. C/EBPβ can activate C/EBPα and PPARγ and produce a cascade reaction that rapidly activates the expression of adipogenesis-related genes [16,17,18,19]; C/EBPα is translated in the early stages of adipocyte differentiation and cooperates with PPARγ to promote adipocyte differentiation [20]. Furthermore, LPL, SREBP1, RXRα, and some members of the KLF family are also involved in the regulation of adipocyte differentiation [10]. ncRNAs, including microRNAs (miRNAs), long non-coding RNAs (lncRNAs) and circular RNAs (circRNAs), have also been shown to be important regulators of adipocyte differentiation in recent years [21,22,23]. For instance, miR-340-5p inhibits ovine adipocyte differentiation by targeting ATF7 [24] and miR-25-3p regulates the differentiation of intramuscular preadipocytes in goats via targeting KLF4 [25]. He et al. (2022) previously constructed an lncRNA-miRNA-mRNA regulatory network during goat intramuscular and subcutaneous adipocyte differentiation by RNA-seq, and identified several lncRNAs that may regulate adipocyte differentiation [26]. Though circRNAs have been reported to affect adipocyte differentiation in cattle [27], pigs [28], and ducks [29], there have been no reports on circRNAs regulating adipocyte differentiation in goats. Therefore, exploring new endogenous regulatory factors is of substantial significance for elucidating the differentiation in goat intramuscular adipocytes.

As a kind of endogenous ncRNA, circRNA widely exists in various cells of various organisms, is abundantly expressed, and has high cell specificity, tissue specificity, and developmental stage specificity [30,31]. CircRNA is produced by the covalent attachment of spliceosome-mediated mRNA 3’ splice sites to 5’ splice sites [32,33]. CircRNA was first recorded in plant viroids in 1976 [34]. According to the origin and generation pattern of genome, circRNA can be divided into three types: intronic circRNAs [35], exonic circRNAs [36], and exon–intron circRNAs [37].CircRNAs have a variety of biological functions, including as a competing endogenous RNA (ceRNA) to sponge miRNA [38], regulate the transcription of its host genes [39], and circRNA also has the potential to encode proteins [40,41,42]. Among them, circRNAs are the most widely studied as ceRNA mechanisms. Based on the multiple biological functions of circRNAs, circRNAs are extensively involved in biological processes such as the occurrence and development of cancer, osteogenic/myogenic/adipogenic differentiation, lipid metabolism, and browning of white adipose tissue [43,44,45,46,47,48]. A large number of studies have shown that circRNAs function as ceRNA to regulate adipocyte differentiation. For instance, in cattle, it has been reported that circFUT10 inhibits adipocyte differentiation via sponging let-7 [45], and circPPARγ facilitates adipocyte differentiation by binding miR-92a-3p and YinYang 1 [49]. However, the effect of circRNAs on goat adipocyte differentiation is still unknown.

In order to identify the circRNAs function in goat adipocyte, the differentially expressed chi_circ_0006511 screened by whole transcriptome sequencing (RNA-seq) of goat intramuscular adipocytes during differentiation was selected as the research object. In this study, we used fluorescence in situ hybridization (FISH), nucleocytoplasmic separation, RNA pull down, overexpression, knockdown, Bodipy and Oil red O staining, dual luciferase report assay and qPCR to explore the effect of chi-circ_0006511 on the differentiation of goat intramuscular adipocytes, and confirmed that the chi-circ_0006511/novel-miR-87/CD36 axis positively regulates GIMPA differentiation, which not only enriched the regulatory network of goat intramuscular adipocyte differentiation but also provides new ideas for further study of the function of fat deposition circRNA. These results provide a basis for exploring the molecular mechanism of intramuscular adipocyte differentiation and serve as a reference for molecular breeding in goats.

## 2. Results

### 2.1. Chi-Circ_0006511 Identification

We obtained 16 differentially expressed circRNAs (data in submission) through the RNA-seq of GIMPA and GIMA (induced for differentiation for 3 days) (Figure 1A). In this study, one of the up-regulated circRNAs, chi_circ_0006511, was selected as the target. chi_circ_0006511 was formed by the circularization of exon 9 and exon 11 of the LMO7 gene on goat chromosome 22, which was a typical exonic circRNA (Figure 1B). Divergent primers containing splicing sites were designed according to the sequence and Convergent primers were designed according to the linear transcript of the chi_circ_0006511 host gene. Using cDNA reverse transcribed from total RNA and linear RNA digested with RNase R as templates, PCR was performed with PrimeSTAR^®^ Max DNA Polymerase, bands were verified by electrophoresis (Figure 1C), and the presence of splice sites was verified by Sanger sequencing (Figure 1B).

### 2.2. Chi-Circ_0006511 Promotes Gimpa Differentiation

We detected the expression of chi-circ_0006511 at different stages of GIMPA differentiation, and the results showed that chi-circ_0006511 was up-regulated at the early stage of differentiation, and then decreased (Figure 2A), suggesting that it has a promoting effect to GIMPA differentiation. Therefore, the constructed pcDNA3.1-circ_0006511-EF1-ZsGreen was transfected as OE group, pcDNA3.1-null-EF1-ZsGreen was transfected as control group (OE-NC), and the transfection efficiency of OE and OE-NC was observed by green fluorescence (Figure 2B). The expression of chi-circ_0006511 increased about 40-fold (Figure 2C), and more lipid droplets accumulation in the OE group was observed by Bodipy and Oil red O staining (Figure 2D), and differentiation marker genes, PPARγ, C/EBPα, C/EBPβ, LPL, SREBP1 were extremely significantly up-regulated (*p* < 0.01) (Figure 2E), indicating that overexpression of chi-circ_0006511 promotes the differentiation of goat intramuscular adipocytes.

To further verify the effect of chi-circ_0006511 on GIMPA differentiation, we used chemically synthesized siRNA to knockdown the expression of chi-circ_0006511 and found that the efficiency of the first siRNA(si1) was about 70% (Figure 3A); therefore, si1 was used for subsequent experiments. We observed that compared with the siNC group, the number of lipid droplets in the si1 group was reduced (Figure 3B) and the differentiation marker genes were extremely significantly down-regulated (*p* < 0.01) (Figure 3C), which was contrary to the result of overexpressing chi-circ_0006511, indicating that chi-circ_0006511 was a positive regulator of GIMPA differentiation.

### 2.3. Chi-Circ_0006511 Sponge Novel-miR-87 as a ceRNA

The intracellular distribution of circRNA is closely related to its function. In order to explore how chi-circ_0006511 regulates GIMPA differentiation, we first identified the subcellular localization of chi-circ_0006511 by nucleocytoplasmic separation and FISH. The results showed that chi-circ_0006511 was distributed in both the nucleus and cytoplasm of goat intramuscular adipocytes (Figure 4A,B), suggesting that it may function as ceRNA. Therefore, we used the circMir, a MiRanda and RNAHybrid-based program to predict its possible sponged miRNAs, obtaining 10 common miRNAs. Theoretically, these miRNAs have opposite expression trends with circRNAs; since chi-circ_0006511 was an up-regulated circRNA in GIMA, we selected 141 down-regulated miRNAs (DEmir-down) in GIMA from our previous RNA-seq data. Combined with circMir results, five common miRNAs were obtained (Figure 4C,D). Then, according to the minimum free energy (mfe) of miRNAs binding with circRNA predicted by RNAHybrid, novel-miR-87 was selected for further study (Figure 4D). Subsequently, the expression of novel-miR-87 after overexpression and knockdown of chi-circ_0006511 were detected, and it was found that chi-circ_0006511 negatively regulates the expression of novel-miR-87(Figure 4E,F), suggesting that it may be a potential target miRNA of chi-circ_0006511. To further verify this speculation, we then designed a biotin-labeled specific probe for chi-circ_0006511 and confirmed that novel-miR-87 was pulled down together with chi-circ_0006511 by RNA pull down (Figure 4G). Finally, the dual-luciferase reporter assay showed that novel-miR-87 mimics inhibited the dual-luciferase activity of psiCHECK2-circ_0006511 WT. In summary, these results indicate that chi-circ_0006511 acts as a ceRNA sponge novel-miR-87.

### 2.4. Novel-miR-87 Inhibits GIMPA Differentiation

What role does novel-miR-87 play in GIMPA differentiation? We synthesized the mimics/inhibitor based on the sequence of novel-miR-87 to mimic/inhibit its expression. The results showed that novel-miR-87 mimics were up-regulated by about 12,000-fold (Figure 5A). At the same time, it was found that lipid droplet accumulation was reduced compared with the mim-NC group (Figure 5B), and adipocyte differentiation marker genes were significantly down-regulated (*p* < 0.01, Figure 5C). As expected, novel-miR-87 inhibitor treatment promoted GIMPA differentiation (Figure 5D–F), indicating that novel-miR-87 was a negative regulator of GIMPA differentiation, which was opposite to the effect of chi-circ_0006511.

### 2.5. Identification of Target Genes of Novel-miR-87

miRNAs mainly regulate the expression of target genes by binding to the 3’UTR region of target genes. Therefore, we used RNAHybrid to predict the target genes of novel-miR-87 and compared them with the target genes of novel-miR-87 predicted by previous transcriptome sequencing. By intersection, 1209 shared genes were obtained (Figure 6A). Then we performed KEGG analysis on 1209 genes and displayed the pathways related to adipocyte differentiation. CD36, which was simultaneously enriched in fat digestion and absorption, the PPARg signaling pathway, adipocytokine signaling pathway and the AMPK signaling pathway, was selected as a potential target gene of novel-miR-87 (Figure 6B). The expression of CD36 was opposite to that of novel-miR-87, novel-miR-87 mimics extremely significantly inhibited CD36 expression (*p* < 0.01, Figure 6C), while novel-miR-87 inhibitor extremely significantly up-regulated CD36 expression (*p* < 0.01, Figure 6D). On the other hand, CD36 was consistent with the expression changes in chi-circ_0006511, and was significantly increased/decreased with overexpression/knockdown of chi-circ_0006511 (*p* < 0.01) (Figure 6E,F). Additionally, novel-miR-87 mimics significantly inhibited the dual-luciferase activity of psiCHECK2-CD36 3’WT (*p* < 0.01, Figure 6G).

### 2.6. Knockdown of CD36 Inhibits GIMPA Differentiation

In order to reveal the effect of CD36 on GIMPA differentiation, we constructed the px459-CD36 vector based on CRISPR/cas9 to knockdown CD36. Firstly, two sgRNAs were designed by CRISPOR (tefor.net) based on the CD36 sequence. The constructed vector was verified by monoclonal PCR and sanger sequencing to confirm that the sgRNA was successfully inserted into the vector (Figure 7A). The successfully constructed vector was then transferred into GIMPA, and the control group (named CD36-KDNC) was transfected with px459 plasmid, CD36-KD1 containing sg1, and CD36-KD2 containing sg2. The expression of CD36 was detected by qPCR and WB (Figure 7B), the results showed that CD36-KD1 had a high knockout efficiency. Compared with CD36-KDNC, less lipid droplets and down-regulated differentiation marker genes was observed in CD36-KD1 (both *p* < 0.01, Figure 7C,D). These results indicated that suppressed CD36 inhibited the differentiation of goat intramuscular adipocytes.

## 3. Discussion

circRNAs had a different splicing feature and special loop structures; the identification of back splicing sites was the basis of circRNAs function verification. For this reason, before elucidating the role of chi_circ_0006511 in regulating GIMPA differentiation, its loop formation needs to be identified. The chi_circ_0006511 back-splice site identified by RNase R digestion, PCR and sanger sequencing was consistent with the sequence by RNA-seq, which proved that the chi_circ_0006511 actually circulated in GIMA.

Based on this, we found that chi_circ_0006511 showed an upward trend from 0 to 84 h of GIMPA differentiation, and a down-regulated trend from 84 to 120 h. It was speculated that chi_circ_0006511 might play a positive role in the early differentiation of GIMPA. Functionally, the overexpression of chi_circ_0006511 promoted lipid accumulation in GIMA, and the mRNA expression levels of adipogenic genes PPARγ, C/EBPα, C/EBPβ, SREBP1, LPL were significantly up-regulated. The upregulation of the expression of these genes suggested that the gain-of-function of chi-circ_0006511 promotes the differentiation of GIMPA. In contrast, knockdown of chi_circ_0006511 by siRNA inhibited lipid accumulation in GIMA, and the mRNA expression levels of adipogenic genes were significantly down-regulated, which was opposite to the trend of overexpression. It could be concluded that chi_circ_0006511 promoted GIMPA differentiation.

Numerous studies have reported that circRNAs act as miRNA sponges to regulate fat deposition in animals. For instance, Jiang et al. (2020) revealed that circFUT10 was abundantly expressed in Qinchuan bovine subadipocytes and circFUT10 combined with let-7c promotes cell proliferation and inhibits cell differentiation by targeting PPARGC1B in bovine adipocytes. Li et al. (2022) reported that circPPARA affects porcine IMF content via adsorbed miR-429 and miR-200b [50]. chi_circ_0006511 belong to exonic circRNA. The composition of circRNAs was closely related to the subcellular localization of circRNAs, which in turn was closely related to the biological functions of circRNAs. Among the many mechanisms of circRNA, the most widely studied and most concerned is the ceRNA mechanism. Exonic RNAs are mostly distributed in the cytoplasm and usually have one or more miRNA binding sites, which can absorb miRNAs by sponge, thereby releasing the inhibitory effect of miRNAs on target genes [51]. Therefore, we first examined the subcellular localization of chi_circ_0006511 and found that chi_circ_0006511 was expressed in both the nucleus and cytoplasm of GIMA, suggesting that it may function as a molecular sponge for miRNAs. By predicting the miRNA bound by chi_circ_0006511 and selecting novel-miR-87 for verification, it was found that chi_circ_0006511 negatively regulates the expression of novel-miR-87, and further confirmed by RNA pull down and dual luciferase reporter assay that chi_circ_0006511 sponged novel-miR-87. Additionally, the inhibitory effect of novel-miR-87 on GIMPA differentiation was verified by gain-of-function and loss-of-function experiments, which was opposite to that of chi_circ_0006511.

miRNAs bind to the 3’ UTR region of target genes, inhibit the expression of target genes, and play an important regulatory role in animal growth and development, cell proliferation, cell differentiation and apoptosis [52,53]. Then, the target genes of novel-miR-87 were predicted by RNAHybrid and RNA-seq, and 1209 common genes were obtained. KEGG analysis of these common genes showed that the PI3K/Akt signaling pathway, adipocytokine signaling pathway, AMPK signaling pathway and other pathways were significantly enriched. Among these, the PI3K/Akt signaling pathway was known to play a central role in cell physiology by mediating growth factor signaling, glucose homeostasis, lipid metabolism, and cell proliferation [54]. In adipocytes, the PI3K/Akt signaling pathway promotes lipid biosynthesis and inhibits lipid hydrolysis and is a positive regulator of adipocyte differentiation [55]. The adipocytokine signaling pathway is the sum of all proteins and factors responsible for regulating adipocytokine, mainly including APN, leptin, IL-6 and TNF-α and so on. APN inhibits adipogenesis through the AMPK pathway, which regulates lipid, cholesterol, and glucose metabolism in specialized metabolic tissues such as liver, muscle, and adipose tissue [56]. CD36 was commonly enriched in the PPARg signaling pathway, adipocytokine signaling pathway and AMPK signaling pathway, and studies have reported that CircScd1 inhibits the formation of lipid droplets in AML-12 hepatocytes through the JAK2/STAT5 pathway, which is achieved by inhibiting the expression of CD36 to block its mediated lipid uptake [57,58,59]. Therefore, we selected CD36 as a potential target gene of novel-miR-87 for further study and verified that novel-miR-87 targeting CD36 3’UTR negatively regulates the expression of CD36. It was also found that CD36 was up-regulated with the overexpression of chi-circ_0006511, and down-regulated with knockdown of chi-circ_0006511.

In order to elucidate the regulatory effect of CD36 on the differentiation of goat intramuscular adipocytes, the goat px459-CD36 vector for knockdown CD36 was constructed, and vectors were transferred to GIMPAs. Oil red O and Bodipy staining showed that knockdown of CD36 inhibited the accumulation of lipid droplets, and the expression levels of adipogenic differentiation marker genes PPARγ, C/EBPα, C/EBPβ and LPL were significantly down-regulated. LPL is mainly expressed in tissues with large amounts of oxidized or stored fatty acids, such as heart, skeletal muscle, brown adipose tissue and white adipose tissue [60]. LPL acts as a scavenging factor lipase [61], can hydrolyze the triglyceride-rich lipoprotein VLDL and triglycerides in chylomicrons. Loss-function of LPL mice were born with significantly elevated blood TG levels and die from inability to absorb butterfat [62], suggesting that LPL is particularly important for fatty acid uptake [63]. Long-chain fatty acids (LCFAs) provide energy for cells, and are also an important component of cell membranes and intracellular lipid storage materials [64]. CD36, a type of membrane protein, is a member of the class B scavenger receptor family and is also a fatty acid transmembrane transporter. CD36 uptakes LCFAs in tissues and promotes lipid accumulation and dyslipidemia [65]. Adipose tissue-specific KO-CD36 mice has reduced cold tolerance because CD36 knockout inhibits lipid accumulation in brown adipose tissue, hinders the uptake of LCFAs in adipose tissue, and increases the content of free LCFAs in blood [66]. As a sensor of LCFAs, CD36 can also activate the PPAR signaling pathway [67] and AMPK signaling pathway [68], while PPARγ can induce CD36 uptake of lipids [69]. The differentiation of GIMA was inhibited with CD36 knockdown, which may be due to the inhibition of the uptake and transport capacity of LCFAs by CD36 and LPL down-regulated, thereby reducing the energy intake during the differentiation of GIMA. At the same time, due to the knockdown of CD36, the activation of the PPARγ and AMPK signaling pathways was inhibited. Our previous research found that CD36 was widely expressed in goat tissues, with the highest expression in adipose tissue, and CD36 expression was positively correlated with goat IMF content, and this study showed that knockdown of CD36 inhibited the differentiation of goat intramuscular adipocytes, indicating that CD36 had a positive regulation on goat IMF deposition. However, although the regulatory role of CD36 in the differentiation of goat intramuscular adipocytes was not yet clear, the specific regulatory mechanism still needs to be further studied.

## 4. Materials and Methods

### 4.1. Cell Isolation, Culture and Induction of Differentiation

Goat intramuscular preadipocytes (GIMPA) were isolated from the longissimus dorsi muscle (Sichuan jianyang dageda animal husbandry CO., LTD, Sichuan, China) of 7-day-old Jianzhou Da-er goats. The isolation and culture of cells was consistent with the previous study [25]. DMEM-F12 (Gibco, Carlsbad, CA, USA) containing 10% FBS (fetal bovine serum, Gibco, USA), 1‰ Penicillin-Streptomycin (Gibco, Carlsbad, CA, USA) and 50 μmol·L^−1^ oleic acid (Sigma, St Louis, MO, USA) was used to induce adipogenic differentiation of GIMPA to goat intramuscular adipocytes (GIMA).

### 4.2. Total RNA Extraction, Rnase R Treatment and cDNA Synthesis

Total RNA was extracted by Trizol reagent (TaKaRa, Tokyo, Japan) according to the instructions. Total RNA (2 μg) was incubated with 3 U/μg of RNase R (Geneseed, Guangzhou, China) for 15 min at 37 °C. After treatment with RNase R, the RNA was equally divided into two parts, one for electrophoresis verification and one for cDNA synthesis by the Revert Aid First Stand cDNA Synthesis Kit (Thermo FisherScientific, Waltham, MA, USA) according to the instructions; the expression levels of chi_circ_0006511 and mRNAs were analyzed by qRT-PCR. Then, 1 μg of untreated RNA was taken as a template directly for reverse transcription; RNA and cDNA were stored at −80 °C and −20 °C, respectively.

### 4.3. Real-Time Quantitative PCR

cDNA was used as the template and reaction system: 10 μL TB GreenTMPremix Ex TaqTMII, 1 μL Foward and reverse primers, 1 μL cDNA, 7 μL ddH_2_O, PCR program: 95 °C for 3 min; 95 °C for 30 s, annealing for 15 s, extension at 72 °C for 15 s, 40 cycles. The primer information is shown in Table 1. qPCR results were processed by the 2^−ΔΔCt^ method, with UXT as the internal reference gene, and the data were expressed as “Mean ± SD”. One-way ANOVA in SPSS 24.0 was used for significance analysis, and Graphpad prism 9.0 was used for drawing.

### 4.4. Fluorescence In Situ Hybridization (FISH)

The specific probe of chi_circ_0006511 conjugated with Cy3 was designed and synthesized by RiboBio CO., LTD (Guangzhou, China). The cellular localization of chi_circ_0006511 was detected by FISH kit (RiboBio, Guangzhou, China). Briefly, the GIMAs cultured with slides in 24 wells were fixed with 4% paraformaldehyde, permeabilized with pre-cooled PBS containing 0.5% Triton X-100 for 10 min, added with prehybridization solution and incubated at 37 °C for 30 min, and then 50 μmol of probe was added to the pre-warmed hybridization solution and incubated with cells at 37 °C overnight. In dark conditions, the slides were washed with 4× (0.1% Tween-20), 2× and 1 × SSC solutions at 42 °C, respectively, and then the slides were fixed on the slides with Mounting Medium with DAPI (abcam, Cambridge, UK). Pictures were taken with a confocal microscope (Zeiss, Oberkochen, Germany).

### 4.5. Nuclear and Cytoplasmic Isolation

Cytoplasmic and nuclear fractions of 1 × 10^7^ GIMAs were isolated by the PARIS™ Kit (AM1556, Thermo Fisher Scientific, Bothell, WA, USA) according to the introduction. Briefly, GIMAs was lysed in Cell Fraction Buffer on ice for 10 min. After centrifugation at 500× *g* for 3 min at 4 °C, the supernatant was collected as a cytoplasmic fraction, followed by washing the pellet with Cell Fraction Buffer. Finally, the nuclei were collected.

### 4.6. Vector Construction

chi_circ_0006511 overexpression vector (pcDNA3.1-circ_0006511-EF1-ZsGreen) and control vector (pcDNA3.1-null-EF1-ZsGreen) were obtained from HanBio (Shanghai, China).

The Crispr/cas9 CD36 vector based on the Px459 for knocking down the expression level of CD36 was constructed following the protocol of Zhang Lab. The sgRNA of CD36 was designed using CRISPOR software, two pairs of sgRNAs with higher scores were annealed and phosphorylated, then the Quick Ligase (NEB, Beijing, China) was used to connect it with the px459 plasmid purified by Bbs I digestion (Thermo Fisher Scientific, WA, USA); after picking a single clone, it was verified by sequencing that the inserted sgRNA was correct.

Goat chi_circ_0006511 wild-type (psiCHECK2- chi_circ_0006511 WT) and mutant (psiCHECK2- chi_circ_0006511 MT) vectors for reporting dual-luciferase activity were constructed by Tsingke (Chengdu, China). CD36 3′UTR was cloned by 3′RACE kit (Takara, Tokyo, Japan), and the CD36 3′ wild-type (psiCHECK2-CD36 3′ WT) and mutant (psiCHECK2-CD36 3′ MT) were constructed by Tsingke (Chengdu, China).

### 4.7. Small RNA Synthesis and Cell Transfection

The interfering RNA of Chi_circ_0006511 was designed and synthesized by HanBio (Shanghai, China) (Table 2). According to the sequence of novel-miR-87, novel-miR-87 mimics and novel-miR-87 inhibitor were designed and synthesized by GemePharma (Shanghai, China) (Table 3). TurboFect™ Transfection Reagent (Invitrogen, Carlsbad, CA, USA) was used to transfect vector, siRNA, miRNA mimics and miRNA inhibitor into 80% confluent GIMPAs, according to the manufacturer’s instructions.

### 4.8. Dual-Luciferase Reporter Assay

The luciferase activity was detected after co-transfection of psiCHECK2-chi_circ_0006511 and psiCHECK2-CD36 3’ with novel-miR-87 mimics, respectively. The dual luciferase reporter system (Vazyme, Nanjing, China) was used to detect the binding relation between chi_circ_0006511 and chi_novel-miR-87, chi_novel-miR-87 and CD36.

### 4.9. Bodipy Staining and Oil Red O Staining

The cells were washed three times by PBS. After discarding the PBS, the cells were fixed with 4% paraformaldehyde for 30 min, and the formaldehyde was discarded and then washed twice with PBS. The Bodipy stock solution (Thermo fisher Scientific, Waltham, MA, USA) was diluted 1:1000 with PBS under dark conditions to obtain the working solution, and 200 μL of the working solution was added to each well for 30 min at room temperature, and the lipids in the goat intramuscular adipocytes were observed by fluorescence microscope.

The Oil red O stock solution (Solarbio, Beijing, China) and ddH_2_O were mixed at a ratio of 3:2 and filtered twice to obtain the Oil red O working solution. Then, 200 μL Oil Red O working solution was added to each well for 30 min at room temperature in the dark for staining, the Oil red O dye was discarded, washed three times with PBS and lipid droplet accumulation was observed and photographed in GIMAs.

### 4.10. Western Blot

The equal amount of protein lysis was separated by 10% SDS-PAGE, and then the protein was transferred to PVDF membrane (Millipore, Billerica, MA, USA), blocked with 5% skim milk at room temperature for 2 h, washed with TBST, incubated with primary antibody overnight at 4°C, and then incubated with secondary antibody. After one hour of incubation at room temperature, the immunoreactive signals were visualized by an ECL kit (BioRad, Hercules, CA, USA). Anti-CD36 for western blot (1:500 dilution, Wanleibio, Shenyang, China), Goat Anti-Rabbit IgG (H + L) HRP for western blot (1:5000 dilution, Abways, Shanghai, China), β-Actin Mouse Monoclonal Antibody for western blot (1:1000 dilution, Abways, Shanghai, China).

### 4.11. RNA Pull Down

RNA pull down was performed to demonstrate the binding between circRNA and miRNA by PureBinding^TM^ RNA-Protein pull-down Kit (Geneseed, Guangzhou, China). Briefly, chi-circ_0006511 specific probes conjugated with biotin (RiboBio, Guangzhou, China) bind to magnetic beads carrying streptavidin, incubated with cell lysate. Then, the magnetic beads were collected using a magnetic frame, and chi-circ_0006511 was pulled down. The obtained total RNA was reverse transcribed using the Mir-X miRNA First-Strand Synthesis Kit (Takara, Tokyo, Japan) and miRNA was detected by qPCR.

## 5. Conclusions

In conclusion, this study clarified the positive regulatory effect of chi-circ_0006511 on GIMPA differentiation and identified mechanistically that chi-circ_0006511 acts through the novel-miR-87/CD36 axis (Figure 8).

## Figures and Tables

**Figure 1 ijms-23-12295-f001:**
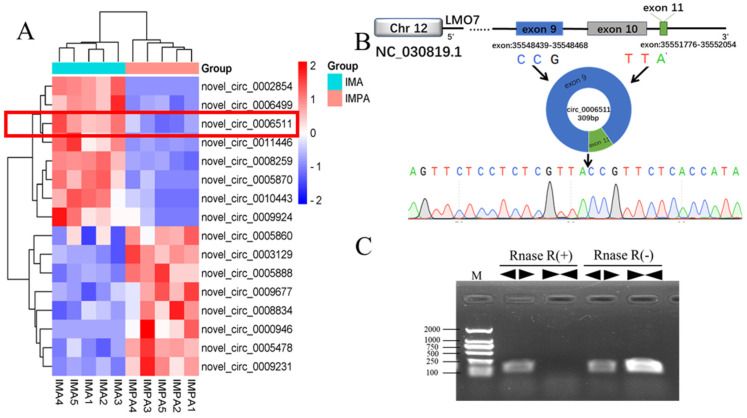
Basic information of chi-circ_0006511. (**A**) Clustering heatmap of differentially expressed circRNAs between GIMPA and GIMA; (**B**)chi_circ_0006511 sequence information; (**C**) Electrophoresis of PCR products of Divergent primer and Convergent primer in Rnase R (+) and Rnase R (−) samples, “
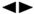
” indicates Divergent primer, “
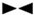
” indicates Convergent primer.

**Figure 2 ijms-23-12295-f002:**
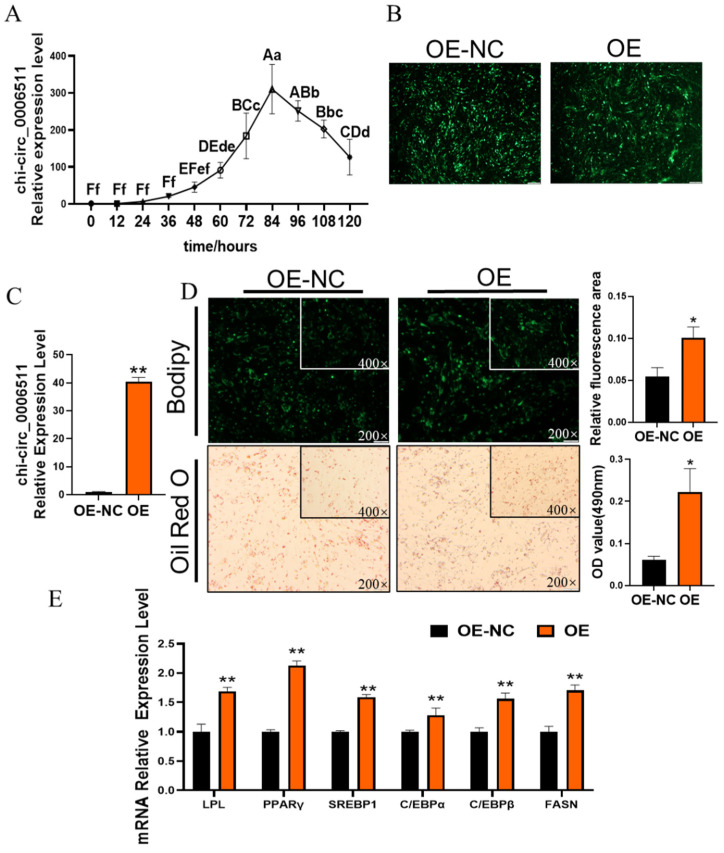
Chi-circ_0006511 overexpression promotes GIMPA differentiation. (**A**) Temporal expression of chi-circ_0006511 during GIMPA differentiation, capital letters indicated extremely significant difference (*p* < 0.01), lowercase letters indicated significant difference (*p* < 0.05); (**B**) Green fluorescent protein (GFP) characterizes the infection efficiency of vectors; (**C**) the overexpression efficiency of chi-circ_0006511. (**D**) The Bodipy and Oil red O staining reveals the lipid droplets in cells, Bodipy fluorescence was quantified with ImageJ; (**E**) the mRNA expression level of adipocytes differentiation related genes. Three biological replicates were set up for each group. Data are expressed as means ± SD. “**” indicated extremely significant difference (*p* < 0.01), “*” indicated significant difference (*p* < 0.05).

**Figure 3 ijms-23-12295-f003:**
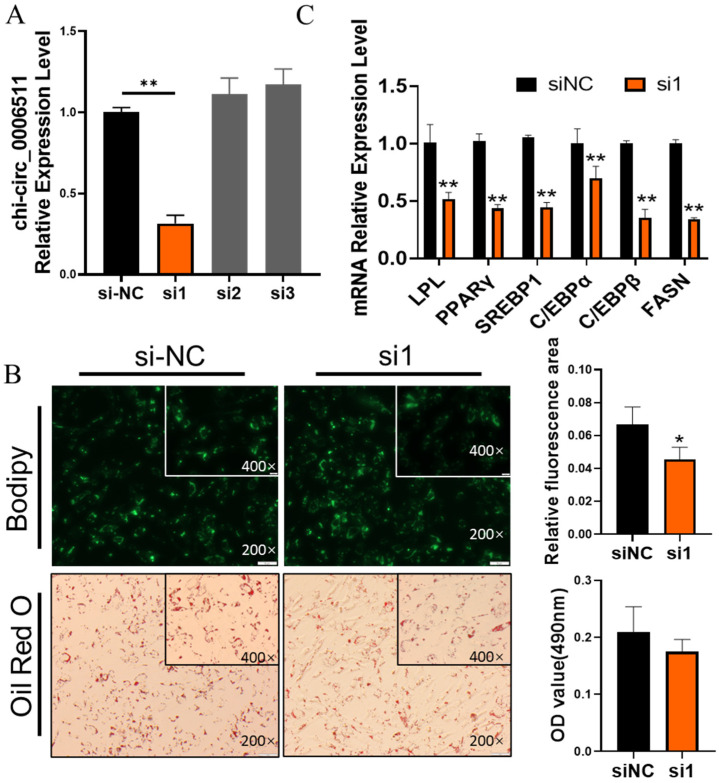
Chi-circ_0006511 knockdown inhibits GIMPA differentiation. (**A**) Efficiency detection of three siRNA knockdown of chi-circ_0006511; (**B**) The Bodipy and Oil red O staining reveals the lipid droplets in cells; (**C**) the mRNA expression level of adipocytes differentiation related marker genes. Data are expressed as means ± SD. “**” indicated extremely significant difference (*p* < 0.01), “*” indicated significant difference (*p* < 0.05).

**Figure 4 ijms-23-12295-f004:**
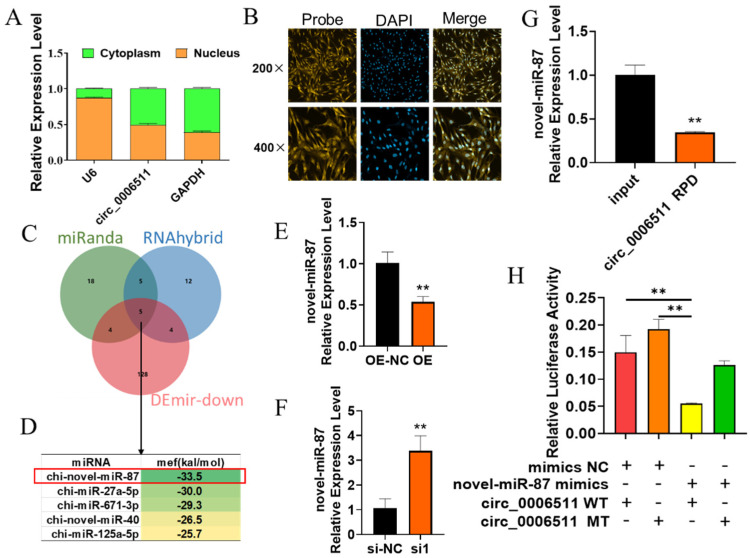
Subcellular localization of chi-circ_0006511 and identification of its targeting relationship with novel-miR-87 as a ceRNA. (**A**) The nucleocytoplasmic localization of chi-circ_0006511 was detected by qPCR, U6 was used as a nuclear reference, and GAPDH was used as a cytoplasmic control to correct the expression level of chi-circ_0006511. (**B**) Subcellular localization detection of chi-circ_0006511 by FISH; (**C**) Venn diagram of chi-circ_0006511 as a ceRNA-bound miRNA prediction by MiRanda, RNAHybrid and coexpression analysis of RNA-seq(Demir-down); (**D**) The Minimum free energy prediction of the intersection of 10 miRNAs binding to chi-circ_0006511 by RNAHyrid; (**E**) novel-miR-87 was down-regulated by chi-circ_0006511 overexpression; (**F**) novel-miR-87 was up-regulated by chi-circ_0006511 knockdown; (**G**) qPCR detected the expression of novel-miR-87 with chi-circ_0006511 pull down assay; (**H**) Dual-luciferase reporter assay to verify the targeting relationship between novel-miR-87 and chi-circ_0006511. Data are expressed as means ± SD. “**” indicated extremely significant difference (*p* < 0.01).

**Figure 5 ijms-23-12295-f005:**
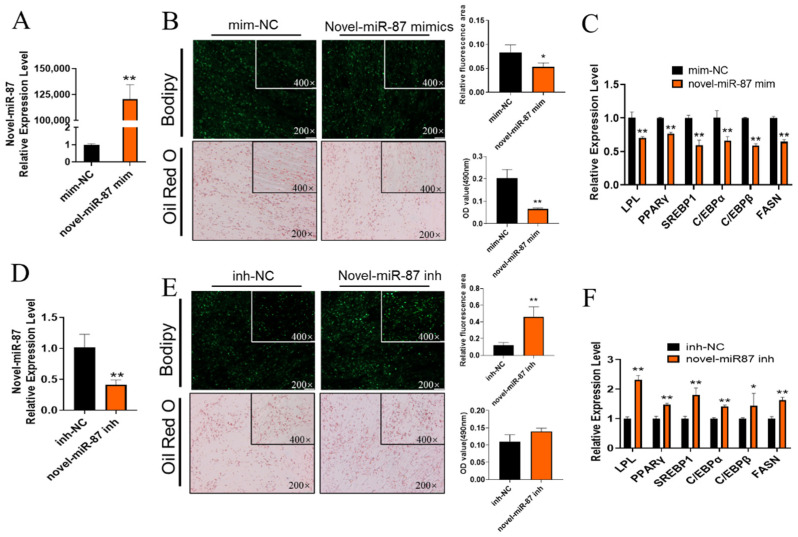
The effect of novel-miR-87 on GIMPA differentiation. Efficiency of novel-miR-87 mimics (**A**) and inhibitor (**D**); The Bodipy and Oil red O staining reveals the lipid droplets novel-miR-87 mimics (**B**) and inhibitor (**E**); (**C**,**F**) the mRNA expression level of adipocytes differentiation related marker genes. Data are expressed as means ± SD. “**” indicated extremely significant difference (*p* < 0.01), “*” indicated significant difference (*p* < 0.05).

**Figure 6 ijms-23-12295-f006:**
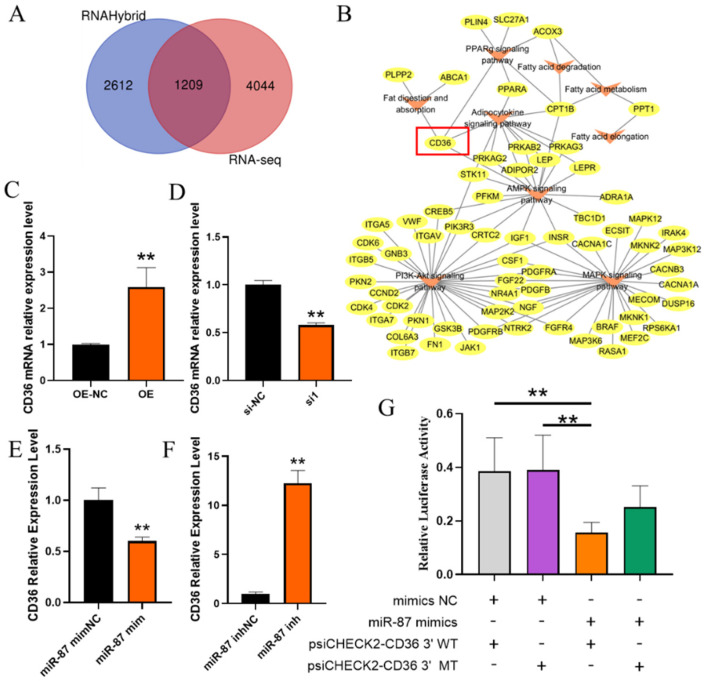
Novel-miR-87 target gene prediction and targeting identification. (**A**) Venn diagram of RNAHybrid and RNA-seq prediction of novel-miR-87 target genes; (**B**) KEGG analysis of intersecting target genes, showed the gene interaction map enriched with adipocyte differentiation-related pathways by Cytoscape; (**C**) CD36 mRNA expression level was up-regulated with overexpression of chi-circ_0006511. (**D**) CD36 mRNA expression level was down-regulated with knockdown of chi-circ_0006511; (**E**) novel-miR-87 mimics down-regulated CD36 mRNA expression; (**F**) novel-miR-87 inhibitor down-regulated CD36 mRNA expression; (**G**) The dual-luciferase reporter gene assay reflected the targeting relationship between novel-miR-87 and CD36. Data are expressed as means ± SD. “**” indicated extremely significant difference (*p* < 0.01).

**Figure 7 ijms-23-12295-f007:**
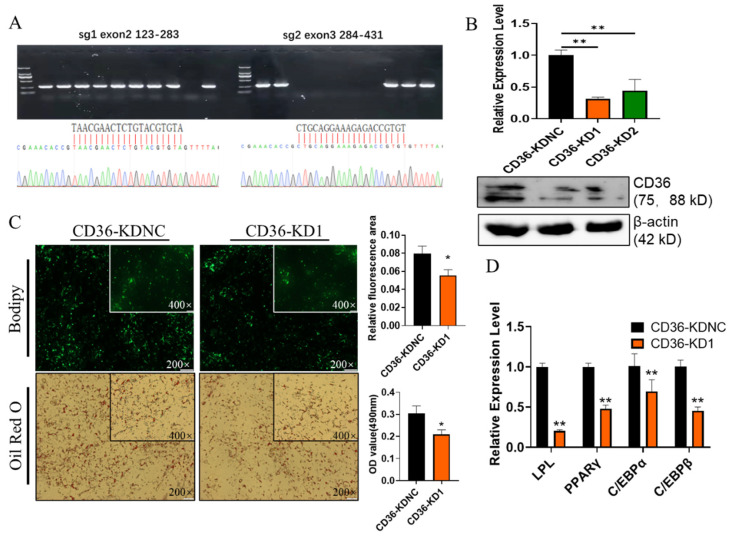
Effects of CD36 on GIMPA differentiation. (**A**) CD36 sgRNA design and monoclonal verification; (**B**) CD36 expression level verification by qPCR and western blot; (**C**) The Bodipy and Oil red O staining reveals the accumulation of the lipid droplets; (**D**) the mRNA expression level of adipocytes differentiation related marker genes. Data are expressed as means ± SD. “**” indicated extremely significant difference (*p* < 0.01), “*” indicated significant difference (*p* < 0.05).

**Figure 8 ijms-23-12295-f008:**
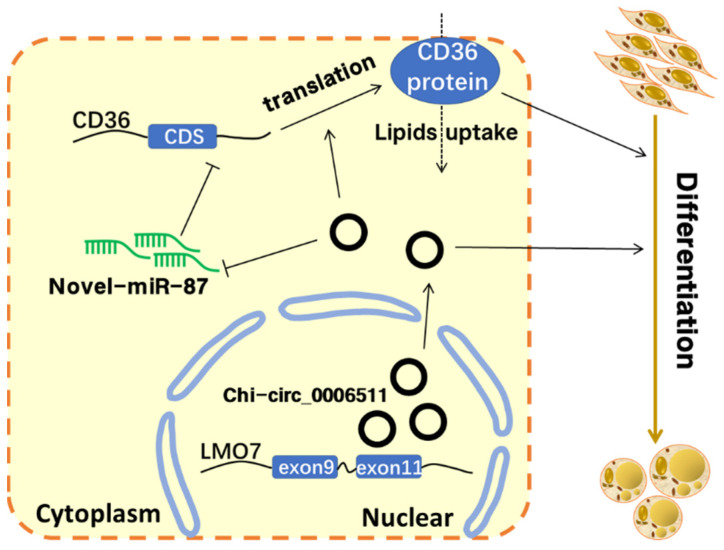
The model of chi-circ_0006511 regulating goat intramuscular adipocyte differentiation. “→” indicate promotion, “⟞” indicates inhibition, and “⇢” indicate that it is in speculation.

**Table 1 ijms-23-12295-t001:** The primer information for PCR or qPCR.

Primer Name	Sequence	Purpose	Products Length	TM/°C
chi_circ_0006511 Divergent	GGATCCCAGAGGATGTGAACTGGA	qPCR	235	60
	TCATCCTTTACAGTGTTGGGAACGC			
chi_circ_0006511 Convergent	GGGCACGAACGTGGAGA	PCR	147	60
	GGGCTTGGCATGACTGG			
Novel-miR-87	ACTAATGGGCTTGGGGAGCCT	Novel-miR-87qPCR		64
A-TY	ATCCAGTGCAGGGTCCGAGG			
CD36	AGAATCCAGACGAAGTGGCA	qPCR	214	60
	ACAGCCAGGTTGAGAATGGT			
SREBP1	AAGTGGTGGGCCTCTCTGA	qPCR	127	58
	GCAGGGGTTTCTCGGACT			
FASN	TGTGCAACTGTGCCCTAG	qPCR	111	57
	GTCCTCTGAGCAGCGTGT			
PPARγ	AAGCGTCAGGGTTCCACTATG	qPCR	197	60
	GAACCTGATGGCGTTATGAGAC			
C/EBPα	CCGTGGACAAGAACAGCAAC	qPCR	142	58
	AGGCGGTCATTGTCACTGGT			
C/EBPβ	CAAGAAGACGGTGGACAAGC	qPCR	204	66
	AACAAGTTCCGCAGGGTG			
LPL	TCCTGGAGTGACGGAATCTGT	qPCR	174	60
	GACAGCCAGTCCACCACGAT			
UXT	GCAAGTGGATTTGGGCTGTAAC	qPCR	180	60
	ATGGAGTCCTTGGTGAGGTTGT			
U6	TGGAACGCTTCACGAATTTGCG	qPCR		60
	GGAACGATACAGAGAAGATTAGC			
px459-U6	GGCCTATTTCCCATGATTCCT	bacterial liquid PCR	267	60

**Table 2 ijms-23-12295-t002:** The sequences of siRNAs for goat chi_circ_0006511-siRNA.

Name	Sequence 5′-3′
Chi-circ_0006511-si1	S: GGUGAGAACGGUAACGAGATT
	A: UCUCCUCUCGUUACCGUUCTT
Chi-circ_0006511-si2	S: GAGAACGGUAACGAGAGGATT
	A: UCCUCUCGUUACCGUUCUCTT
Chi-circ_0006511-si3	S: GAACGGUAACGAGAGGAGATT
	A: UCUCCUCUCGUUACCGUUCTT
siNC	S: UUCUCCGAACGUGUCACGUTT
	A: ACGUGACACGUUCGGAGAATT

**Table 3 ijms-23-12295-t003:** The sequences of mimics and inhibitor for goat novel-miR-87.

Name	Sequence 5′-3′
miR-87 mimNC	S: UUCUCCGAACGUGUCACGUTT
	A: ACGUGACACGUUCGGAGAATT
miR-87 mim	S: UGGGCUUGGGGAGCCUGGGACU
	A: UCCCAGGCUCCCCAAGCCCAUU
miR-87 inhNC	CAGUACUUUUGUGUAGUACAA
miR-87 inh	AGUCCCAGGCUCCCCAAGCCCA

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
