# Peer review of "Chi-Circ_0006511 Positively Regulates the Differentiation of Goat Intramuscular Adipocytes via Novel-miR-87/CD36 Axis"

_ijms, 2022, doi:10.3390/ijms232012295_

Round 1

Reviewer 1 Report

In this manuscript, Li et al., showed that the circRNA 0006511 enhances intramuscular preadipocyte differentiation illustrated by enhanced lipid accumulation and increased mRNA expression of differentiation marker genes. Its mechanism of action is attributed to sequestering miR-87 that acts on the CD36 target mRNA.

1.     While the authors were able to elucidate a circRNA-miRNA-mRNA access, it is unclear what role this pathway plays in vivo.

2.     Also, amongst all the differentially expressed circRNAs why was 0006511 picked for further analysis. What other miRNAs does circRNA 0006511 regulate?

3.     Please explain the bioinformatic methods that used to find mRNA targets of miRNAs in greater detail. What are the limitations of RNAHybrid? Wouldn’t a more straightforward way be to find mRNAs that are differentially regulated in 0006511 OE or KD? Or perform a CLIP experiment for miR-87 to show direct association?

4.     The mechanism of action of CD36 in intramuscular adipocyte differentiation is unclear. This is detrimental to the significance of this study.

5.     Are these circRNAs, miRNAs conserved across species? Shouldn’t the authors look for something specific to goat intramuscular adipocytes that explains the high quality of goat meat?

6.     The significance of this study is unclear. Can the authors elaborate on why this is important to study?

7.     Can the authors comment why the decrease in lipid accumulation by Bodipy or Oil Red O is so modest in circRNA knockdown, CD36 knockdown, and miR-87 mimics.

8. The authors need to significantly improve their english writing and proofreading of the manuscript. The entire abstract does not make sense. 

Reviewer 2 Report

This manuscript provided by Li et al. identified the positive regulatory effect of chi-circ_0006511 on GIMPA differentiation in goats, and discovered that chi-circ_0006511 acts through the novel-miR-87/CD36 axis. The results are solid and strongly support the conclusions. In general, it is well written and the experiments are well designed. I would suggest the publication on IJMS after minor revisions to improve the quality of this manuscript.

My major concern is the quality of the figures. Many of the figures are not professionally prepared.

1. Please make the "A/B/C..." the same font throughout all the figures. 

2. Make sure the text (titles of all the charts...) in all the figures with same font, at least make them clearly visible. For example, Figure 2A and D, the x- and y-axis titles are too small to read. 

3. Rearrange the size of the charts. In figure 5, 5A x-axis is extended to 5B.

4. Figure 6B, I suggest to point out the CD36 so that readers can easily find its position in the pathway and catch your point.

5. Make the text in the last figure (conclusion) bigger to read.

Round 2

Reviewer 1 Report

The authors made some changes that were asked for. The significance of this study is still lacking. It is unclear why circRNAs versus mRNAs are studied in the process of adipogenic differentiation? It is also unclear how this circRNA can be repurposed to benefit goat meat production in a real-life setting. Can the authors site examples of its translational potential? The abstract is still poorly written and doesn't capture interest or convey significance. The authors should provide more background on circRNAs and discuss why they should be studied as opposed to other mechanisms that control mRNA expression. 

Reviewer 2 Report

I am satisfied with the authors' response and my concerns are well addressed. I suggest the publication of this manuscript on IJMS.

Author Response

Thank you for your recognition of this work, your suggestion is helpful to us, thank you!